# The Effect of E-Health Literacy and Patient–Physician Relationship on Treatment Adherence

**DOI:** 10.3390/healthcare13060632

**Published:** 2025-03-14

**Authors:** M. Kemal Boz, Mesut Çimen

**Affiliations:** Department of Healthcare Management, Institute of Health Science, Acibadem Mehmet Ali Aydinlar University, İstanbul 34638, Türkiye

**Keywords:** e-health literacy, treatment adherence, patient–physician relationship

## Abstract

**Objective:** The Internet is actively used for gathering and learning health-related knowledge. The uncertainty surrounding the reliability of health information found online has heightened the importance of effective relationships between patients and physicians, as well as the value of a physician’s guidance. This study focused on investigating the impact of individuals’ e-health literacy level and their relationship with their physicians on their treatment adherence. In addition, the aim was to determine the mediatory role of patient–physician relationship in the effect of e-health literacy on treatment adherence. **Materials and Methods:** The study population included volunteering participants over the age of 18 living in the city of Istanbul. The study employed a survey method, gathering data from 425 participants between 15 April and 15 August 2023. **Results:** According to the results of the research, e-health literacy has an effect on patient–physician relationships, patient–physician relationship has an effect on treatment adherence, and e-health literacy has an effect on treatment adherence. It was found that patient–physician relationship played a mediatory role in the relationship between e-health literacy and treatment adherence. **Conclusions:** E-health literacy and patient–physician relationship were found to be factors affecting treatment adherence. Some suggestions were made within the framework of these results.

## 1. Introduction

With the shift from an industrial society to an information society, the meanings attributed to health and disease, behaviors in health and disease states, as well as mutual relations and relationship patterns between the physician and the patient have changed; new social support networks have emerged, and medical knowledge has become globalized [1]. Information, which was previously typically obtained through written sources, can be obtained from digital media due to intensive Internet usage. Especially with the widespread use of smartphones, the virtual world has been made available to us as an unlimited gold mine of information that can be accessed at any time. According to the global digital headlines 2023 report, 5.44 billion people worldwide use smartphones, 5.16 billion individuals use the Internet, and 4.76 billion are active on social media platforms [2]. A study conducted by the Turkish Statistical Institute (TurkStat) revealed that the rate of household access to the Internet in Turkey will reach 95.5% in 2023 [3]. In the same way as other sectors, the healthcare field could not turn a blind eye to the rapid technological developments. Based on the Eurostat 2022 report, 52% of individuals in European Union member states seek health-related information online. This rate is also above 50% in Turkey. The top health-related inquiries concern injuries, diseases, nutrition, and improving health [4]. 

Access to healthcare services is now easier thanks to the widespread use of digital health technologies, yet certain people now face additional obstacles. Access to digital health services is influenced by a number of factors, including geographic location, age, education level, and socioeconomic status. People who lack e-health literacy might not be able to make full use of health services, which could harm the patient–physician relationship and make treatment compliance challenging.

However, inequalities in health directly affect the effective use of digital health resources. Individuals with a high ability to obtain information from digital health platforms can be more successful in their own health management, while those who lack this skill may experience more difficulties in their treatment processes. A key factor in lessening the effects of these discrepancies is patient–physician communication. Physicians can improve their patient’s health literacy and help them stick to their treatment plans by teaching them how to use digital health resources.

However, the dissemination of false information on digital health platforms is another significant element that has a major impact on adherence to treatment and the patient–physician relationship. Inaccurate health information that circulates on social media and the Internet might cause patients to doubt medical advice and make poor judgments. People with limited e-health literacy are particularly in danger from this. It is imperative that medical practitioners educate their patients about the dangers of false information and refer them to trustworthy online resources.

In summary, health issues including the digital divide, health disparities, and access to false information are all closely correlated with e-health literacy. Among the key tactics to improve treatment adherence are enhancing the patient–physician interaction, educating people about the proper use of digital health tools, and sharing trustworthy health information.

## 2. Study Model and Hypotheses

The study model consists of the path mediation model with the mediatory role of patient–physician relationship in the relationship between e-health literacy and treatment adherence (Figure 1).

### 2.1. E-Health Literacy

Health literacy (HL) was first discussed as a term by Scott Simonds in 1974 [5]. The World Health Organization has defined HL as “the ability of individuals to possess the cognitive and social skills required to access, comprehend, and utilize health information for the purpose of enhancing and maintaining their health” [6]. Currently, the convenience of Internet access and the ability to obtain the desired information quickly and abundantly have affected the healthcare sector, and the notion of e-health literacy has arisen. Individuals who access information about their health conditions, diseases, and treatment methods through the Internet have started to have a say in their health status. Expanded access to information has facilitated the global sharing of medical knowledge and caused diseases or patients that were previously unknown to surface on social media [1].

E-health literacy refers to the capacity to locate, comprehend, assess, and utilize health-related information from digital platforms to tackle or solve health issues [7]. The notion of e-health literacy acts as a measure of an individual’s competence to seek, recognize, and evaluate the information required to make informed health choices and to implement the knowledge they have acquired in their daily lives [8]. Increased e-health literacy levels of individuals have positive effects both on themselves and their environment. E-health applications facilitate quick access to health resources, and their advantages include improving health, reducing costs, making the right decisions on health-related issues, and creating a healthier society. Adoption of positive health behaviors also reduces possible medical errors [9].

Although the Internet is used extensively for collecting information and learning about health, research shows that the information is not based on scientific data, the updated reference sources are questionable, and expert opinions are not included on a majority of the websites. Consequently, the uncertain reliability of health-related information available online makes patient–physician relationships essential. This is because if individuals come across false information regarding their health status on the Internet and believe that this information is accurate, this may affect their treatment processes negatively and may even lead to faulty diagnosis, treatment, and health behaviors [10].

### 2.2. Patient–Physician Relationship

The patient–physician relationship is based on mutual support as well as mutual respect and trust. This complicated relationship is affected by many factors, and physicians have a great role in managing the patient–physician relationship. Physicians should be able to determine their approach to each patient individually and prioritize the patient’s interests during the treatment process. The physician, who works to enhance their patient’s life satisfaction, elevate the patient’s health status, and make the patients lead a healthy lives, must convince their patients that they share the same goal with the patient. Conversely, the patient should also participate in the treatment process, take an active role in the decisions made, and try to communicate well with their physician. In the studies conducted, it was observed that the participants had the opinion, “I would talk to a doctor or other healthcare personnel about the health-related information accessed on the internet”. Accordingly, it may be said that patients are influenced by the Internet and communicate with healthcare personnel to share the information they obtain from the Internet [11].

It is considered that 70% of the diagnoses made in healthcare provision processes occur as a result of a good patient–physician dialogue [12]. A correctly established relationship increases patient satisfaction and increases the likelihood of positive results from medical practices [13,14]. Therefore, the saying “diseases are not unique, patients are unique” may also be interpreted as determining the type of relationship according to the patient instead of establishing the same relationship model for each patient [15].

### 2.3. Treatment Adherence

Adherence refers to the degree to which a person’s actions align with the health-related instructions or recommendations provided by a healthcare professional concerning a specific disease or condition [16]. It includes not delaying any follow-up examinations to be made by the physician, complying with the treatment plan, using the medicines prescribed by the physician appropriately, and changing the behaviors that the physician requested to be changed. Treatment nonadherence, which is defined as the opposite, occurs in the form of improper or no use of medications and the patient leaving the treatment process unfinished, and is most commonly observed in patients receiving psychiatric treatment [17]. Treatment nonadherence increases the burden on healthcare institutions, the number of readmissions for treatment, morbidity and mortality rates, and patients also experience problems with their quality of life [18].

In the studies conducted, it has been observed that individuals mainly obtain health-related information from the Internet, find this information useful, and trust the information, and when they experience health issues, they first search for their complaints on the Internet but never start or discontinue their treatments based on the information they obtain from the Internet despite all these behaviors. It was also found that during the treatment process with their physicians, they generally searched for information about their treatment on the Internet and believed that the information they accessed from the Internet would help them improve their health [19]. 

### 2.4. Patient-Centered Care

The Institute of Medicine’s “Crossing the Quality Chasm” study from 2001 marked the introduction of the phrase “patient-centered care” into the language of health policy. “Knowing and respecting patients’ values, preferences, and needs, promoting partnership with patients in their decisions, ensuring patients’ physical and emotional comfort, and advocating for patients” is how this report characterized patient-centered care [20].

Additionally, according to the International Alliance of Patient Organizations (IAPO), “Patients should be at the center of the health care system and therefore the system should be designed around the patient” [21].

The patient’s ability to make educated decisions is the cornerstone of patient-centered care. In order to determine and satisfy the patient’s requirements and preferences, healthcare professionals should consult with the patient and their family members while making decisions.

According to all of these definitions, patient-centered care encourages patients to actively participate in creating new care models and choosing their own course of treatment. Personal preferences, desires, and situations in all facets of their lives are taken into consideration when making decisions, in addition to clinical results and physical health [22]. Thus, a method of providing healthcare that prioritizes the patient in all decision-making and treatment procedures is known as patient-centered care [23]. In conclusion, the relationship between the patient and the healthcare provider who participates in decision-making processes is the foundation of patient-centered care.

## 3. Method

### 3.1. Sample

This research employed the survey method, a quantitative technique, for data collection. The study has a cross-sectional design. A total of 425 participants from Istanbul contributed data between 15 April and 15 August 2023. The collected data are related to demographic characteristics, e-health literacy, patient–physician relationship, and treatment adherence. E-health literacy was evaluated through the scale designed by Norman and Skinner (2015), patient–physician relationship was measured using the scale developed by Maly and colleagues (1998), and treatment adherence was measured using the scale developed by Hausman and colleagues (2001).

The following were determined as criteria for inclusion in this study: living in Istanbul and being over 18 years of age is considered sufficient. No exclusion criteria were set based on the educational status of the participants. The sample used in the study represents 0.0027 (27 per thousand) of the general population of Istanbul. A total of 217 of the participants were male, and 208 were female. A total of 20 people under the age of 20, 314 people between the ages of 20 and 40, and 91 people over the age of 40 participated. Of the participants, 23 were primary school graduates, 124 were high school graduates, 227 were undergraduate and associate degree graduates, 41 were master’s degree graduates, and 10 were doctoral graduates.

The surveys were distributed to respondents who have different socio-demographic characteristics to reduce the risk of bias. Four hundred fifty (450) participants in the sample who are over 18 years old, have different occupations and social classes, and live in Istanbul were selected randomly and invited to participate in the survey, and all accepted to participate in the study. However, 25 out of the 450 filled surveys that were found to be incomplete, incorrect, or irrelevant were excluded from the analyses. The higher response rate might be due to fact that the number of questions was not too many to answer, and the average time to complete the whole survey was only 2 min.

### 3.2. Ethical Approval

Before starting the study, the ethical approval decision numbered 2023-2/39 and dated 27 January 2023 was taken from Acıbadem Mehmet Ali Aydınlar University Medical Research Evaluation Board (ATADEK) indicating that the study was appropriate in terms of medical ethics. Participation in the sample was entirely voluntary, and all individuals were asked to read the informed consent form before joining the study. All individuals gave their informed consent in written form.

### 3.3. Scales

The study’s data collection survey is structured into four distinct parts. The initial section comprises the demographic information form in which the participants’ gender, age, and education level, as well as the importance of the Internet and the significance of accessing health resources for them when making health-related decisions, are investigated. In the second section of the questionnaire, there is a scale form consisting of statements to measure e-health literacy. E-Health Literacy (eHEALS: The Ehealth Literacy Scale) was developed by Norman and Skinner in 2006. The scale consists of eight items. The scale was adapted into Turkish by Gencer (2016). Reliability was calculated with Cronbach’s Alpha method since the method used in the e-health literacy scale was a Likert-type measurement. The calculated alpha value of 0.915 indicates a high degree of reliability. The third part of the questionnaire consists of statements about the patient–physician relationship. The patient–physician relationship was measured with the 10-item “Perceived Efficacy in Patient-Physician Interactions scale” developed by Maly et al. (1998) [24]. Ünal et al. translated and adapted the scale into Turkish in 2018. The Cronbach’s Alpha value of the patient–physician relationship scale was found to be 0.905, which indicates high reliability. The fourth section of the questionnaire includes statements related to treatment adherence. As a treatment adherence scale, the 5-item scale developed by Hausman et al. (2001) [25] and translated and adapted into Turkish by Deniz et al. (2021) [26] was used. The Cronbach’s Alpha value of the scale was 0.832 for the “Adherence Scale”. The scale has sufficient conditions for reliability.

### 3.4. Statistical Analysis

Data analysis in this study was conducted with the statistical programs SPSS 21.0 and AMOS 22.0. Confirmatory factor analysis (CFA), item-total correlation, and Cronbach’s Alpha were employed to assess the validity and reliability of the scales. After conducting the validity and reliability analyses, scale scores were summarized in a descriptive statistics table, including mean, standard deviation, skewness, and kurtosis values. Skewness and kurtosis coefficients are considered in the normality test of the score. When assessing the normal distribution of scores obtained from a continuous variable, if the skewness and kurtosis coefficients lie within the ±1 range, it indicates that the scores do not significantly deviate from a normal distribution. The scores consisting of the average, standard deviation, skewness, and kurtosis values obtained as a result of reliability and validity analyses of the scale structures are presented in the descriptive statistics table. The Pearson correlation test was conducted to analyze the relationship between the scale scores. In the study, a mediation model was developed to evaluate the mediating role of the patient–physician relationship in the connection between the independent variable (e-health literacy) and the dependent variable (treatment adherence). The significance level (*p*) for the analyses was set at 0.05.

## 4. Results

### 4.1. Respondents’ Characteristics

Of the 425 participants, 48.9% were female and 51.1% were male. A total of 51.3% of the participants were aged 20–29, 27.3% were aged 30–39, 15.3% were aged 40–49, 4.7% were aged 50–59, and 1.4% were aged 60 and above. Of the participants, 5.4% had primary school, 29.2% had high school, 24.7% had associate degrees, 28.7% had undergraduate, and 12% had graduate levels of education. In relation to the usefulness of the Internet for health decision-making, 3.3% of the participants stated that it was not useful at all, 13.4% stated that it was not useful, 16% had no opinion, 56.7% stated that it was useful, and 10.6% stated that it was very useful. Regarding the significance of accessing health resources, 3.1% of the participants stated that it was not important at all, 9.2% stated that it was not important, 13.6% had no opinion, 54.4% stated that it was important, and 19.8% stated that it was very important (Table 1).

According to Table 2, the e-health literacy score was 3.78 ± 0.84, the patient–physician relationship perception score was 4.09 ± 0.72, and the treatment adherence score was 4.17 ± 0.78. The participants’ e-health literacy level, patient–physician relationship perception, and treatment adherence scores were high.

### 4.2. The Relationship Between E-Health Literacy, Patient–Physician Relationship and Treatment Adherence

According to Table 3, there was a positive and moderate correlation between e-health literacy and patient–physician relationship (r = 0.49; *p* < 0.05) and treatment adherence (r = 0.39; *p* < 0.05), and there was a positive and significant relationship between patient–physician relationship and treatment adherence (r = 0.60; *p* < 0.05). Moreover, according to the results in Table 3, patient–physician relationship plays a stronger role in treatment adherence than e-health literacy.

### 4.3. The Effect of E-Health Literacy on Treatment Adherence and the Mediatory Role of Patient–Physician Relationship

Structural equation modeling was used to estimate the effects of e-health literacy and patient–physician relationship on treatment adherence and the mediating effect of e-health literacy on treatment adherence through patient–physician relationship.

E-health literacy has an effect on patient–physician relationships. (β = 0.54; t = 8.53; *p* < 0.05).

Patient–physician relationship has an effect on treatment adherence (β = 0.81; t = 11.17; *p* < 0.05).

E-health literacy has a moderate effect on treatment adherence (β = 0.47; t = 8.34; *p* < 0.05).

Although the mediation of the patient–physician relationship in the relationship between e-health literacy and treatment adherence approaches significance, it does not meet the standard threshold for statistical significance (Table 4).

In line with the analyses, four hypotheses could be accepted to be true.

## 5. Discussion

In this research, the effect of individuals’ e-health literacy level and their relationships with their physicians on their adherence to treatment was examined.

Based on the findings of the research conducted by Akbolat et al., functional literacy had a positive effect on patient–physician relationship; Inoue et al. found that communicative literacy was linked to the patient–physician relationship, and Schwartzberg et al. found that there was a strong correlation between a low level of functional health literacy and ineffective relationship [27,28,29].

According to the results of this study, the relationship between the patient and the physician affects patient adherence to treatment more than e-health literacy. These results are similar to some studies in the literature. Deniz et al. concluded that the relationship between the patient and the physician affects patient adherence to treatment [26]. Orom et al. stated that the quality of the patient–physician relationship improves adherence and positively affects treatment outcomes [30]. Schmidt claims that the patient–physician relationship strengthens the patient’s adherence to any treatment [31]. In another study, Akbolat et al. suggest that the relationship between patient and physician based on trust and accurate communication increases patient adherence to treatment [32]. Dopelt et al. came to the conclusion in their study that favorable outcomes in health services are caused by physicians who demonstrate empathy, give patients clear and understandable information, and build a relationship of trust with their patients [33]. In their study, Peimani et al. looked at patients with type 2 diabetes and discovered that when they had good patient–physician contact, people with higher e-health literacy demonstrated improved self-care behaviors and better treatment adherence [34]. The results of all these studies reveal the importance of the patient–physician relationship in patient adherence to treatment.

E-health literacy has a moderate effect on treatment adherence. In studies conducted on the subject, similar results were obtained with our findings. In their study, Çavuşoğlu et al. determined that the most important variables predicting adherence to treatment during the pandemic were regular blood pressure monitoring, having COVID-19, and health literacy [35]. In another study, Dopelt et al. concluded that individuals with chronic diseases who have high e-health literacy tend to monitor and manage their diseases more effectively, are better able to overcome adversities they may encounter, and respond faster to changes [36]. In their study, Green et al. found that dialysis patients with low health literacy were more likely to skip dialysis sessions and visit the emergency department [37]. In another study, Bakan et al. found a positive relationship between health literacy and treatment adherence in hypertension patients [38]. In a study by Miller examining health literacy and adherence to medical treatment in acute and chronic diseases, it was stated that patients who adhered to treatment had better health literacy [39]. Wu et al. discovered that e-health literacy and self-efficacy are crucial to patients’ disease management procedures and improve treatment adherence in their study of older adults with chronic non-communicable diseases [40]. Patients with better e-health literacy were more likely to actively seek out health information, connect with their doctors more successfully, and adhere to recommended treatments, according to Lu et al.’s study on the function of e-health literacy in online health communities [41]. Health literacy treatments were successful in enhancing the health of people with chronic illnesses, according to Shao et al.’s study, Health literacy interventions among patients with chronic diseases: A meta-analysis of randomized controlled trials [42]. Higher e-health literacy levels were associated with greater pharmaceutical treatment adherence, according to Aslan et al.’s study [43].

As a result of this study, it was determined that mediation of the patient–physician relationship in the relationship between e-health literacy and treatment adherence approaches significance, it does not meet the standard threshold for statistical significance. The relationship between the patient and the physician plays an important role both in the diagnosis and treatment of diseases and in treatment adherence. When patients’ e-health literacy levels are high, their treatment adherence behaviors increase, while the positive relationships they establish with their physicians also contribute to this process. When the literature is examined, found that patients with low health literacy experienced communication difficulties [44,45]. In their study, Heijmans et al. found that patients with inadequate health literacy gave lower scores to hospital-based communication [46]. According to Refahi et al., treatment adherence and health outcomes can be positively impacted by a high level of e-health literacy. However, it has also been shown that certain patient groups may suffer as a result of unequal access to digital health technologies [47]. 

The study’s findings demonstrate that e-health literacy improves the patient–physician connection, which in turn boosts treatment adherence. In order to improve the patient–physician interaction, it is crucial to create digital health literacy initiatives. Fighting false information and improving the accuracy of the data shown on digital health platforms are also essential. As digital health technologies become more widely used, there is a growing need for health literacy training programs to enable people to utilize these tools with awareness. Enhancing e-health literacy can help people access health services more intelligently, which can improve patient–physician relationships.

Reducing health disparities and expanding access to healthcare services depend heavily on e-health literacy. Thus, training materials should be created specifically for underprivileged groups, and public health campaigns should incorporate digital health education programs. To encourage e-health literacy, awareness-raising initiatives, including public service announcements, social media campaigns, and community-based educational initiatives, should be carried out. Both individuals and medical professionals should receive e-health literacy training as part of public health campaigns. Physicians and other healthcare providers should help their patients obtain trustworthy health information and help them use the appropriate digital health services.

## 6. Conclusions

The results imply that adherence to treatment is significantly influenced by the patient–physician relationship. By using patient-centered communication strategies in clinical practice, physicians can improve their patients’ comprehension and utilization of health information. Patients should receive recommendations from physicians for reliable digital health platforms and instructions on how to utilize them properly. To enhance treatment adherence, physicians and other healthcare providers can also determine their patients’ degree of e-health literacy and offer tailored health counseling. Programs for teaching digital health should be extended within the parameters of public health regulations. By concentrating on those with low health literacy, these initiatives can lessen disparities in information availability.

There should be more initiatives aimed at populations that have barriers to healthcare access because of the digital divide. Municipalities, ministries of health, and community health clinics can arrange in-person training for people with a poor Internet connection. To stop false information from spreading on e-health platforms, public authorities should take steps to highlight trustworthy digital health resources. It is possible to arrange e-health literacy classes to provide people with the tools they need to handle their own medical records. Effective use of health applications, selecting trustworthy sources, and communicating with physicians should all be covered in these sessions. Hospitals, family health clinics, and pharmacies can employ information screens, brochures, and brief training videos to help patients become more e-health literate. When it comes to managing chronic diseases, patient education is crucial. By consciously using e-health technologies, people with conditions like COPD, diabetes, or hypertension can better adjust to medication.

To investigate the long-term effects of e-health literacy on treatment adherence and the patient–physician relationship, longitudinal cohort studies can be carried out. The effects of telemedicine platforms, AI-enabled healthcare, and mobile health apps on patient engagement over time should be studied. The same people can be periodically followed up with to look at behavioral changes in order to evaluate the long-term effects of education initiatives or awareness-raising activities.

Studies examining the effects of digital health literacy on patient–physician relationships and treatment adherence in nations with disparate health systems can be carried out within the context of e-health literacy and patient involvement in various nations. Understanding health disparities may be improved by conducting research contrasting the effects of digital health literacy in industrialized and poor nations.

The efficacy of intervention programs (online courses, practical training, etc.) to raise patient groups’ digital health literacy can be evaluated within the parameters of intervention-based studies. To evaluate the impact of AI-supported health coaches, smartphone apps, and digital health platforms on patient engagement, randomized controlled trials can be carried out. It is possible to evaluate algorithms or verification systems that limit patients’ exposure to erroneous health information.

The findings of this study revealed a positive and significant relationship between the participants’ e-health literacy, their patient–physician relationship, and treatment adherence. Within the framework of these results, several recommendations were made to increase the e-health literacy levels of individuals, improve patient–physician relationships, and increase treatment adherence. In order to improve the patient–physician relationship, it is recommended that physicians allocate more time to their patients, that physicians listen to their patients more carefully and sympathetically, that patients are sufficiently informed about the treatment process, and that systems are put in place where patients can easily reach their physicians in case of need. It is also recommended that physicians should be kind, patient, and honest in their relationship with patients and their relatives, prioritize personal privacy, and act in the best interest of the patient. It is important that all healthcare professionals provide support for society to obtain reliable and evidence-based health-related information. Regarding e-health literacy, the health and science literacy of society should be increased, inquisitive thinking should be adopted, and the necessity of investigating the accuracy of the information obtained before applying it should be explained with examples. Treatment adherence is among the most important factors for a treatment to be successful. Therefore, recommendations should be made in accordance with the values, beliefs, and lifestyle of the patient. It is very important for the physician to clearly explain the possible side effects of medications that may be encountered during the treatment process, especially in a language that patients can understand. It is a state policy to keep drug prices under control considering the low-income groups, to provide health assistance to those in need, to include treatment processes and scientifically valid publications and evidence on physicians’ websites, to establish an individual physician follow-up system for the elderly, and to plan face-to-face meetings between physicians and patients at certain intervals will help improve treatment adherence. It is thought that it would be beneficial to undertake similar research with more participants or in different geographic settings.

## 7. Limitations

There are several restrictions on this study. First, the results may not be as broadly applicable as they may be because the study was limited to a specific population and was carried out in Istanbul. Furthermore, it could be challenging to completely evaluate the influence of various confounding factors due to the absence of comparability across demographic characteristics. The questionnaire was given to participants with varying sociodemographic traits in order to lower the possibility of bias in the research. We invited people over the age of 18 from a variety of socioeconomic and occupational backgrounds in an effort to broaden the sample’s diversity.

The study’s technique of gathering data was self-report questionnaires. Due to the potential for memory loss, social preference bias, or subjective impressions, this could result in self-report bias. To reduce this bias, objective measurement techniques (such as clinical evaluations or third-party observations) are advised for future studies. Furthermore, a more thorough evaluation of the current findings might result from research examining the effects of demographic variables using a larger and more representative sample.

## Figures and Tables

**Figure 1 healthcare-13-00632-f001:**
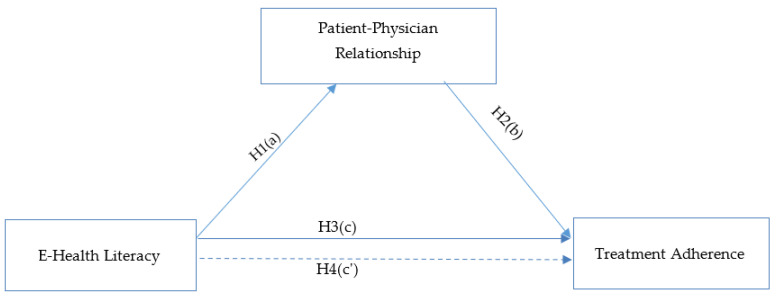
Study model-1. H_1_: E-health literacy has an effect on patient–physician relationships. H_2_: Patient–physician relationship has an effect on treatment adherence. H_3_: E-health literacy has an effect on treatment adherence. H_4_: Patient–physician relationship has a mediating role in the relationship between e-health literacy and treatment adherence.

**Table 1 healthcare-13-00632-t001:** Socio-demographic characteristics of the participants.

Demographic Variable	Groups	*n*	%
Gender	Female	208	48.9
	Male	217	51.1
Age groups	20–29 years	218	51.3
	30–39 years	116	27.3
	40–49 years	65	15.3
	50–59 years	20	4.7
	60 years and older	6	1.4
Education status	Primary education	23	5.4
	High School	124	29.2
	Associate degree	105	24.7
	Undergraduate	122	28.7
	Graduate	51	12.0
Usefulness of the Internet in making health-related decisions	Not useful at all	14	3.3
	Not useful	57	13.4
	No opinion	68	16.0
	Useful	241	56.7
	Very useful	45	10.6
Importance of access to health resources	Not important at all	13	3.1
	Not important	39	9.2
	No opinion	58	13.6
	Important	231	54.4
	Very important	84	19.8

Participants’ e-health literacy, patient–physician relationship, and treatment adherence scores.

**Table 2 healthcare-13-00632-t002:** Descriptive statistics of participants’ e-health literacy, patient–physician relationship, and treatment adherence scores.

Scale and Dimension	*n*	Min.	Max.	$\bar{X}$	SD	Skewness	Kurtosis
E-Health Literacy	425	1.00	5.00	3.78	0.84	0.21	−0.17
Patient–Physician Relationship	425	1.00	5.00	4.09	0.72	−0.05	−0.49
Treatment Adherence	425	1.00	5.00	4.17	0.78	−0.26	−0.65

**Table 3 healthcare-13-00632-t003:** The relationship between e-health literacy, patient–physician relationship, and treatment adherence scores.

Variable	E-Health Literacy	Patient–Physician Relationship	Treatment Adherence
E-Health Literacy	1	0.490 **	0.386 **
Patient–Physician Relationship		1	0.602 **
Treatment Adherence			1

** *p* < 0.01.

**Table 4 healthcare-13-00632-t004:** The effect of e-health literacy on treatment adherence and the mediatory role of patient–physician relationship.

	Independent Variable	Path	Dependent Variable	H	Β	T	*p*	R^2^
**Independent Models**	E-Health Literacy	→	Patient–Physician Relationship	H1 (a)	0.54	8.53	0.000	0.296
	X^2^/sd = 3.043 SRMR = 0.046 GFI = 0.915 NNFI = 0.946 CFI = 0.958 RMSEA = 0.069							
	Patient–Physician Relationship	→	Treatment Adherence	H2 (b)	0.81	11.17	0.000	0.657
	X^2^/sd = 3.675 SRMR = 0.040 GFI = 0.918 NNFI = 0.939 CFI = 0.954 RMSEA = 0.079							
	E-Health Literacy	→	Treatment Adherence	H3 (c)	0.47	8.34	0.000	0.219
	X^2^/sd = 2.317 SRMR = 0.026 GFI = 0.956 NNFI = 0.978 CFI = 0.984 RMSEA = 0.056							
**Model with Mediating Variables**	**Independent** **Variable**	**Path**	**Dependent** **Variable**	**H**	**Mediating** **Variable**	**IE**	**SBT**	**R^2^_EB_**
	E-Health Literacy	→	Treatment Adherence	H4 (c′)	Patient–Physician Relationship	0.448	6.862	0.065
	X^2^/sd = 2.594; SRMR = 0.041; GFI = 0.900; NNFI = 0.948; CFI = 0.957; RMSEA = 0.061							

IE: indirect effect size, SBT: Sobel test statistic, R^2^_IE_: variance due to indirect effect.

## Data Availability

Data are contained within the article.

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
