# Peer review of "The Effect of E-Health Literacy and Patient–Physician Relationship on Treatment Adherence"

_healthcare, 2025, doi:10.3390/healthcare13060632_

Round 1
Reviewer 1 Report
Comments and Suggestions for Authors
The publication titled "The Effect of E-Health Literacy and Patient–Physician Relationship on Treatment Adherence" presents a study about the effects of e-health literacy. The study design was described along with the selected methodology and the results.
The main question addresses the impact of individuals’ e-health literacy level and their relationship with their physicians on their treatment adherence.
The topic is an active research question and part of patient empowerment. This research addresses the relationship between patient and physicians, which is underrepresented in the patient empowerment research so far.
This research confirms findings in published materials specifically that e-health literacy influences patient-physician relationship, patient-physician relationship influences treatment adherence, e-health literacy influences treatment adherence.
The authors used a survey method for the actual study. Combining it with other methods e.g. interviews or focus groups would allow more insights from patients.
The conclusions are consistent with the evidence and arguments presented, the method is presented in section 5 and provides insight in the followed approach, section 6 provides the results of the study stating that the majority of participating subject use the internet to get information about their disease. These findings correlate to a good relationship between patient and physician. This is in summary stated in the conclusion section.
- The text in figure 1 is cut off
- For the completeness of your publication it would be useful, if you would add the questionnaires that you used e.g. in the appendix
Author Response
Dear reviewer, first of all, thank you very much for your time and for your valuable comments and contributions to my manuscript. I have made the necessary corrections and additions and I am submitting my article to you again for your consideration. Kind regards.
Comments 1: The text in figure 1 is cut off.
Response 1: The cut text in Figure 1 has been corrected. (Page 3)
Comments 2: For the completeness of your publication it would be useful, if you would add the questionnaires that you used e.g. in the appendix.
Response 2: The questionnaire used for the research is attached as an appendix at the end of the manuscript. (Page 26)

Reviewer 2 Report
Comments and Suggestions for Authors
Thank you so much for this opportunity to review the paper
Please, below few comments you might consider or clarify:
There are a few overstatements in the manuscript that need revision:
Strength of Correlation (Table 3)
The correlation between e-health literacy and patient–physician relationship (r = 0.49) is moderate, not strong.
The manuscript describes it as a strong relationship, which is an overstatement.
Revision Needed: Please adjust the language to "moderate correlation" rather than "strong relationship."
Main Predictor of Treatment Adherence (Table 3)
The patient–physician relationship is the strongest predictor of adherence (r = 0.60), yet the discussion focuses more on e-health literacy.
Revision Needed: Please highlight that patient–physician relationship plays a stronger role in treatment adherence than e-health literacy.
Regression Analysis (Table 4)
The direct effect of e-health literacy on adherence (β = 0.47) is described as strong, but it is moderate in strength.
Revision Needed: The manuscript should acknowledge this as a "moderate" effect rather than a "strong" effect.
Mediation Analysis (Table 4)
The mediation effect (β = 0.448, p = 0.065) is not statistically significant at p < 0.05.
The manuscript incorrectly claims significance for mediation.
Revision Needed: Reword the mediation discussion to state that the mediation approaches significance but does not meet the standard threshold for statistical significance.
Acknowledge study limitations (sampling bias, confounders, self-report bias)
Update references to include more recent literature (2023–2024).
Best wishes
Author Response
Dear reviewer, first of all, thank you very much for your time and valuable comments and contributions to my article. I have made the revisions you mentioned, and I have also specified the limitations of my study in a broader way. I have added 7 recent studies to my references. I present it to you again for your valuable comments. Sincerely.
Comments 1: The correlation between e-health literacy and patient–physician relationship (r = 0.49) is moderate, not strong. The manuscript describes it as a strong relationship, which is an overstatement. Revision Needed: Please adjust the language to "moderate correlation" rather than "strong relationship."
Response 1: According to Table 3, there was a positive and moderate correlation between e-health literacy and patient-physician relationship (r = 0.49; p < 0.05) and treatment adherence (r = 0.39; p < 0.05), and there was a positive and significant relationship between patient-physician relationship and treatment adherence (r = 0.60; p < 0.05). (Page 10).
Comments 2: The patient–physician relationship is the strongest predictor of adherence (r = 0.60), yet the discussion focuses more on e-health literacy. Revision Needed: Please highlight that patient–physician relationship plays a stronger role in treatment adherence than e-health literacy.
Response 2: Moreover, according to the results in Table 3, patient-physician relationship plays a stronger role in treatment adherence than e-literacy. (Page 10).
Comments 3: The direct effect of e-health literacy on adherence (β = 0.47) is described as strong, but it is moderate in strength. Revision Needed: The manuscript should acknowledge this as a "moderate" effect rather than a "strong" effect.
Response 3: E-health literacy has a moderate effect on treatment adherence (β = 0.47; t = 8.34; p < 0.05). (Page 11).
Comments 4: The mediation effect (β = 0.448, p = 0.065) is not statistically significant at p < 0.05. The manuscript incorrectly claims significance for mediation. Revision Needed: Reword the mediation discussion to state that the mediation approaches significance but does not meet the standard threshold for statistical significance.
Response 4: Although the mediation of the patient-physician relationship in the relationship between e-health literacy and treatment adherence approaches significance, it does not meet the standard threshold for statistical significance. (Page 11).
Comments 5: Acknowledge study limitations (sampling bias, confounders, self-report bias).
Response 5: There are several restrictions on this study. First, the results may not be as broadly applicable as they may be because the study was limited to a specific population and was carried out in Istanbul. Furthermore, it could be challenging to completely evaluate the influence of various confounding factors due to the absence of comparability across demographic characteristics. The questionnaire was given to participants with varying sociodemographic traits in order to lower the possibility of bias in the research. We invited people over the age of 18 from a variety of socioeconomic and occupational backgrounds in an effort to broaden the sample's diversity.
The study's technique of gathering data was self-report questionnaires. Due to the potential for memory loss, social preference bias, or subjective impressions, this could result in self-report bias. To reduce this bias, objective measurement techniques (such as clinical evaluations or third-party observations) are advised for future studies. Furthermore, a more thorough evaluation of the current findings might result from research examining the effects of demographic variables using a larger and more representative sample. (Page 15).
Comments 6: Update references to include more recent literature (2023–2024).
Response 6: Refahi H, Klein M, Feigerlova E. E-Health Literacy Skills in People with Chronic Diseases and What Do the Measurements Tell Us: A Scoping Review. Telemedicine and e-Health, 2023; Vol. 29, No. 2. (Page 12).
Wu Y, Wen J, Wang X, Wang Q, Wang W, Wang X, Xie J, Cong L. Associations Between E-Health Literacy and Chronic Disease Self-Management in Older Chinese Patients With Chronic Non-Communicable Diseases: A Mediation Analysis. Public Health, 2022; Volume 22, article number 2226. (Page 12).
Dopelt K, Bachner Y.G, Urkin J, Yahav Z, Davidovitch N, Barach P. Perceptions of Practicing Physicians and Members of the Public on the Attributes of a “Good Doctor”. Healthcare, 2022, 10, 73. (Page 11).
Peimani M, Stewart A.L, Ghodssi-Ghassemabadi R, Nasli-Esfahani E, Ostovar A. The moderating role of e-health literacy and patient-physician communication in the relationship between online diabetes information-seeking behavior and self-care practices among individuals with type 2 diabetes. BMC Primary Care,2024; Volume 25, Article number: 442 (Page 11).
Lu X, Zhang R. Association Between eHealth Literacy in Online Health Communities and Patient Adherence: Cross-sectional Questionnaire Study. J Med Internet Res. 2021; Sep 13;23(9) (Page 12).
Shao Y, Hu H, Liang Y, Hong Y, Yu Y, Liu C, Xu Y. Health literacy interventions among patients with chronic diseases: A meta-analysis of randomized controlled trials. Patient Education and Counseling, 2023; Volume 114. (Page 12).
Aslan G, Kant E. Relationship Between Medication Adherence and E-Health Literacy Levels in Patients with Hypertension. Turk J Cardiovasc Nurs 2024; 15(36):1-7. (Page 12).

Reviewer 3 Report
Comments and Suggestions for Authors
The study aims to examine the impact of e-health literacy and the patient-physician relationship on treatment adherence. Additionally, it explores the mediating role of the patient-physician relationship in the connection between e-health literacy and adherence to treatment.
Some suggestions to improve it:
Introduction:
- The introduction would benefit from a more detailed discussion of the theoretical frameworks underlying the study. For example, incorporating models of health literacy, patient-centered care, or adherence behavior could provide a stronger conceptual foundation.
2. The study model and hypotheses section should be moved to the end of the introduction.
3. Connect the study to broader healthcare challenges, such as the digital divide, health disparities, and the spread of misinformation in digital health sources.
Methods:
1. Specify whether the study follows a cross-sectional design.
2. Provide more details on sampling. Explain how representative the sample is of the Istanbul general population.
Results:
- Add sub-headings.
- Clearly state whether full mediation or partial mediation was observed in the relationship between e-health literacy, patient-physician relationships, and treatment adherence.
Discussion:
- Cite more studies to strengthen the discussion. For example, https://doi.org/10.3390/healthcare10010073
- Emphasize the practical implications of the results. Explain how the study’s findings can be used to improve healthcare communication and promote digital health education.
- Consider policy implications, such as incorporating e-health literacy training into public health initiatives.
Conclusion:
- Clearly outline how the findings can be applied in clinical practice, public health, and patient education.
- Reiterate the study’s limitations, such as generalizability (due to the sample being from Istanbul) and the cross-sectional nature of the research.
- Suggest areas for future research, including longitudinal studies, cross-cultural comparisons, or intervention-based studies to test strategies for improving digital health literacy and patient adherence.
Author Response
Dear reviewer, first of all, thank you very much for your time, valuable comments and contributions to my article. I have made the revisions you indicated for all sections and I am submitting it to you again for your valuable comments. Kind regards.
Comments 1: The introduction would benefit from a more detailed discussion of the theoretical frameworks underlying the study. For example, incorporating models of health literacy, patient-centered care, or adherence behavior could provide a stronger conceptual foundation.
Response 1: Patient-centered care has been added to the introduction. (Page 5-6).
Comments 2: The study model and hypotheses section should be moved to the end of the introduction.
Response 2: The study model and hypotheses section has been moved to the end of the introduction. (Page 3).
Comments 3: Connect the study to broader healthcare challenges, such as the digital divide, health disparities, and the spread of misinformation in digital health sources.
Response 3: The study is linked to broader health care issues. (Page 2-3).
Comments 4: Specify whether the study follows a cross-sectional design.
Response 4: The study has a cross-sectional design. It is indicated in the sample section. (Page 6).
Comments 5: Provide more details on sampling. Explain how representative the sample is of the Istanbul general population.
Response 5: A detailed explanation is given in the sample section. (Page 6-7).
Comments 6: Add sub-headings.
Response 6: Sub-headings have been added to the results section. (Page 8-9-10).
Comments 7: Clearly state whether full mediation or partial mediation was observed in the relationship between e-health literacy, patient-physician relationships, and treatment adherence.
Response 7: Although the mediation of the patient-physician relationship in the relationship between e-health literacy and treatment adherence approaches significance, it does not meet the standard threshold for statistical significance. (Page 11).
Comments 8: Cite more studies to strengthen the discussion. For example, https://doi.org/10.3390/healthcare10010073
Response 8: In addition, 7 more recent studies are cited in the discussion section.
Refahi H, Klein M, Feigerlova E. E-Health Literacy Skills in People with Chronic Diseases and What Do the Measurements Tell Us: A Scoping Review. Telemedicine and e-Health, 2023; Vol. 29, No. 2. (Page 12).
Wu Y, Wen J, Wang X, Wang Q, Wang W, Wang X, Xie J, Cong L. Associations Between E-Health Literacy and Chronic Disease Self-Management in Older Chinese Patients With Chronic Non-Communicable Diseases: A Mediation Analysis. Public Health, 2022; Volume 22, article number 2226. (Page 12).
Dopelt K, Bachner Y.G, Urkin J, Yahav Z, Davidovitch N, Barach P. Perceptions of Practicing Physicians and Members of the Public on the Attributes of a “Good Doctor”. Healthcare, 2022, 10, 73. (Page 11).
Peimani M, Stewart A.L, Ghodssi-Ghassemabadi R, Nasli-Esfahani E, Ostovar A. The moderating role of e-health literacy and patient-physician communication in the relationship between online diabetes information-seeking behavior and self-care practices among individuals with type 2 diabetes. BMC Primary Care,2024; Volume 25, Article number: 442 (Page 11).
Lu X, Zhang R. Association Between eHealth Literacy in Online Health Communities and Patient Adherence: Cross-sectional Questionnaire Study. J Med Internet Res. 2021; Sep 13;23(9) (Page 12).
Shao Y, Hu H, Liang Y, Hong Y, Yu Y, Liu C, Xu Y. Health literacy interventions among patients with chronic diseases: A meta-analysis of randomized controlled trials. Patient Education and Counseling, 2023; Volume 114. (Page 12).
Aslan G, Kant E. Relationship Between Medication Adherence and E-Health Literacy Levels in Patients with Hypertension. Turk J Cardiovasc Nurs 2024; 15(36):1-7. (Page 12).
Comments 9: Emphasize the practical implications of the results. Explain how the study’s findings can be used to improve healthcare communication and promote digital health education.
Response 9: It is added that the study results can be used to improve health communication and promote digital health education. (Page 12-13).
Comments 10: Consider policy implications, such as incorporating e-health literacy training into public health initiatives.
Response 10: Recommendations were made for the inclusion of e-health literacy training in public health initiatives. (Page 13).
Comments 11: Clearly outline how the findings can be applied in clinical practice, public health, and patient education.
Response 11: Areas of application of the study findings are indicated. (Page 13-14)
Comments 12: Reiterate the study’s limitations, such as generalizability (due to the sample being from Istanbul) and the cross-sectional nature of the research.
Response 12: The generalizability and limitations of the study were reiterated. (Page 15).
Comments 13: Suggest areas for future research, including longitudinal studies, cross-cultural comparisons, or intervention-based studies to test strategies for improving digital health literacy and patient adherence.
Response 13: Research areas to improve digital health literacy and patient adherence were proposed. (Page 14).

Round 2
Reviewer 2 Report
Comments and Suggestions for Authors
Thank you for addressing the comments
Reviewer 3 Report
Comments and Suggestions for Authors
The paper can be accepted in its present form.